# A 4-DOF Upper Limb Exoskeleton for Physical Assistance: Design, Modeling, Control and Performance Evaluation

**Muhammad Ahsan Gull** [1,*], **Mikkel Thoegersen** [2], **Stefan Hein Bengtson** [3], **Mostafa Mohammadi** [2], **Lotte N. S. Andreasen Struijk** [2], **Thomas B. Moeslund** [3], **Thomas Bak** [4] and **Shaoping Bai** [1,*]

1   Department of Materials and Production, Aalborg University, 9220 Aalborg, Denmark
2   Rehabilitation Engineering and Robotics, Center for Rehabilitation Robotics, Department of Health Science and Technology, Aalborg University, 9220 Aalborg, Denmark; mt@hst.aau.dk (M.T.); mostafa@hst.aau.dk (M.M.); naja@hst.aau.dk (L.N.S.A.S.)
3   Department of Architecture and Media Technology, Aalborg University, 9000 Aalborg, Denmark; shbe@create.aau.dk (S.H.B.); tbm@create.aau.dk (T.B.M.)
4   Department of Electronics Systems, Aalborg University, 9220 Aalborg, Denmark; tba@es.aau.dk
*   Correspondence: mag@mp.aau.dk (M.A.G.); shb@mp.aau.dk (S.B.); Tel.: +45-81931727 (M.A.G.)

**Abstract:** Wheelchair mounted upper limb exoskeletons offer an alternative way to support disabled individuals in their activities of daily living (ADL). Key challenges in exoskeleton technology include innovative mechanical design and implementation of a control method that can assure a safe and comfortable interaction between the human upper limb and exoskeleton. In this article, we present a mechanical design of a four degrees of freedom (DOF) wheelchair mounted upper limb exoskeleton. The design takes advantage of non-backdrivable mechanism that can hold the output position without energy consumption and provide assistance to the completely paralyzed users. Moreover, a PD-based trajectory tracking control is implemented to enhance the performance of human exoskeleton system for two different tasks. Preliminary results are provided to show the effectiveness and reliability of using the proposed design for physically disabled people.

**Keywords:** wheelchair upper limb exoskeleton robot; ADL assistance; PD control; dynamic modeling of an upper limb exoskeleton; trajectory tracking; wearable exoskeleton

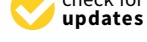

## 1. Introduction

Cervical spinal cord injury (SCI) may result in incomplete or complete tetraplegia and lead to paralysis of all four extremities. Upper limb onset is one of the most profound impairments that significantly degrades the life of individuals with tetraplegia by compromising independence and social interactions. Moreover, it imposes a substantial financial burden on society in the long run. While advanced medical and surgical techniques, such as stem cell therapy, nerve transfer surgery, etc., have been used to restore the upper limb functionality, in some severe cases, it is hard to achieve desired results. Emerging technologies, such as assistive robots, can provide an alternative way to facilitate individuals with physical impairments in activities of daily living (ADL) [1,2] or therapeutic exercises [3,4].

During the past few decades, upper limb exoskeletons used for power amplification and rehabilitation have attracted intensive attention from the health care and engineering sectors [5]. However, given the utility and growing demand of exoskeletons for physical assistance, the technology still faces challenges in mechanical design, controls, and human–robot interaction. Of them, the mechanical design of a shoulder exoskeleton, including kinematic and kinetic analysis, is a major issue in developing an ergonomic system [6]. Christensen et al. [7] proposed a new three degrees of freedom (DOF) spherical mechanism to comply with the human glenohumeral joint movements. The proposed mechanism takes advantage of the double parallelogram (DPL) mechanism, which connects two revolute joints to achieve a spherical workspace and maintains a remote center of motion (RCM).

The results from the biomechanical analysis of the DPL mechanism presented in [7] have shown its significance for the exoskeleton applications [1]. Similarly, Castro et al. [8] presented a novel 3-DOF curved scissor mechanism that connects two revolute joints. The proposed mechanism complies with the human shoulder movements by maintaining the instantaneous center of rotation. Since the above mechanisms can support complex shoulder movements and provide a singularity-free workspace, the passive internal rotation has made it difficult to use for individual with tetraplegia. Alternatively, several other designs, including fully active or hybrid mechanisms to comply with the shoulder anatomical movements, were proposed [9–12]. These exoskeletons support the full range of shoulder girdle movement by preserving the remote center of rotation, but their effects on supporting the physically impaired people in common ADL have not yet been evaluated [13]. Moreover, flexible and parallel mechanisms have also been investigated to reduce inertial problem, but their size and complexity remain issues to be further addressed. Apart from the shoulder exoskeletons, exoskeletons that can support human forearm [4,14–17] and wrist movements [18–20] were developed. Among the existing mechanisms, a direct drive method and a C-ring mechanism are commonly used to support human forearm extension/flexion movements and wrist rotation, as reported in [2,5,20,21].

The feasibility of using an upper limb exoskeleton cannot only be proved by its design. Selection of a control method for improved physical human–robot interaction (pHRI) is essential for successful implementation and user acceptance. Regarding the trajectory tracking problem, proportional-derivative (PD) and proportional-integral-derivative (PID) controllers have been widely investigated for the different types of exoskeletons. Ease of implementation without having prior knowledge of robot dynamics and an ability to independently tune the control parameters have made the PD/PID control method among the most widely used control schemes [22]. However, in the PID controller, an integrator usually reduces the bandwidth of a closed loop system and removes the steady state error caused by extensive disturbance and uncertainties. Alternatively, a high value of the integrator gain may compromise the transient performance and destroys the system's stability. Therefore, many robotic manipulators, including exoskeletons, use purely PD control or PD control with relatively small integral gain [1,22–24]. It is known that a PD controller can guarantee a semi global asymptotic stability after appropriately tuning the gains [23,24].

Several studies have been conducted to modify the linear PID controller that can guarantee an asymptotic stability. For example, PD control with sliding mode compensation [25], PD-based fuzzy sliding mode control [1], PD control with neural compensation [22] and so on. It is well understood that the PD controller can guarantee the stability for the robotic manipulators, but the asymptotic stability cannot be achieved if the robot dynamic contains gravitational torque. The exoskeleton presented in this study is designed to safely support the user in their ADL, especially the C-ring mechanism designed for shoulder and wrist rotation and the worm gear used to drive the elbow joint exoskeleton to hold the output position without energy consumption because of its non-backdrivability [26]. Moreover, hard constraints in the joint mechanisms may not allow the users to move beyond the safety limits.

In this paper, we present a PD control in the joint space to control the four degrees of freedom (DOF) upper limb exoskeleton robot [2] and investigate its effect as an assistive device to support individuals with physical impairments of the upper limbs in a set of ADLs. The contribution of the article can be summarized as follows.

1.  The proposed design can support the human upper limb musculoskeletal structure in basic ADL by providing a kinematically safe and singularity-free workspace. The deign along with the PD control is able to provide a satisfactory tracking performance. It is hypothesized that the trajectory tracking for C-ring mechanism and worm gear mechanism is less prone to the variation in payload, weight of human arm, and exoskeleton due to its ability to hold the output position without energy consumption.

2.  The integration of the upper limb exoskeleton with the CarbonHand glove (BioServo Technologies AB, Kista, Sweden) offers a new paradigm that not only supports the user in manipulation but facilitates them also in hand opening and closing. The experimental evaluation has shown that the proposed design with the PD control scheme is appropriate in performing several ADLs, such as eating/drinking.

The paper is organized as follows. The mechanical design of a wheelchair exoskeleton is presented in Section 2 together with the kinematic modeling required to fulfil the task requirements in operational space. The dynamic model of the upper limb exoskeleton along with the PD control scheme is presented in Section 3. Moreover, the PD controller implementation along with the experimental results on the wheelchair exoskeletons is illustrated in Section 4. Subsequently, a discussion on the exoskeleton performance and its potential future directions are presented in Section 5. The work is finally concluded in Section 6.

## 2. Upper Limb Exoskeleton Robot

### 2.1. Mechanical Design

This section presents a design of an adaptive 4-DOF upper limb wheelchair mounted exoskeleton that can actively support the wearer in performing their activities of daily living, such as eating and drinking. The exoskeleton was designed after carefully analyzing the human upper limb biomechanics. To reduce the complexity of the human biomechanics, several studies have modeled the human arm as 7 degrees of freedom kinematics system by enforcing the simplifications to the upper limb joints and segments [27]. However, we have noticed that the 4-DOF exoskeleton is sufficient for the most common ADLs and keeps the workspace of the human upper extremity intact.

The exoskeleton in Figure 1b is designed as an open-chain structure to replicate the anatomy of human right upper limb and provides a controllable assistive torque to each joint. To describe the design and complete functioning of a robotic exoskeleton, we have separated the design into three sub-modules, i.e., shoulder joint mechanism, elbow module and a wrist module.

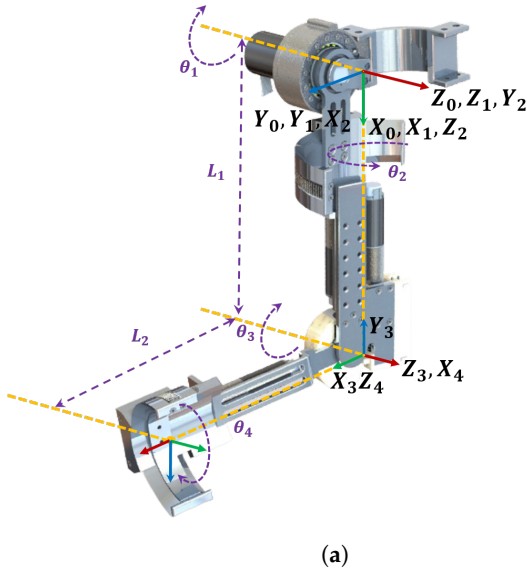
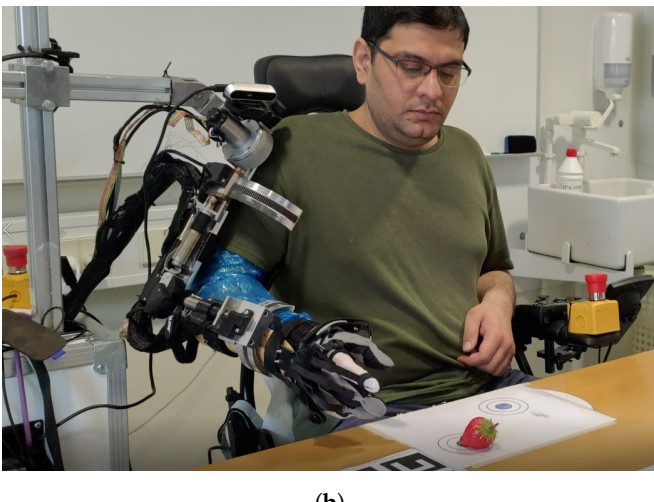

(**a**)  (**b**)

**Figure 1.** Overview of a 4-DOF wheelchair exoskeleton. (**a**) Mechanical model of 4-DOF upper limb exoskeleton. (**b**) A prototype of a wheel chair exoskeleton with carbon hand developed from SEM glove (Supplementary Materials).

The human shoulder (Glenohumeral) joint is modeled as a 3-DOF spherical joint that describes the orientation of the human upper arm. These three successive rotations are abduction/adduction, extension/flexion, and internal/external rotation. Hence, an open chain serial mechanism with three revolute joints whose axes of rotation intersect

at a common point is kinematically equivalent to a spherical joint. Based on this observation, we have designed a shoulder mechanism that can actively support the 2-DOF glenohumeral joint movements such as shoulder extension/flexion movement and shoulder internal/external rotation, as shown in Figure 1a. The shoulder abduction/adduction movement is passively adjustable. Locking the upper arm abduction movement will prevent the user from moving beyond the wheelchair workspace, causing uncomfortable interaction with an external environment. The complete design of the shoulder mechanism, shown in Figure 1a, is able to preserve the dynamic center of rotation throughout its workspace. The exoskeleton's extension/flexion is achieved by a direct drive brushless DC motor (EC-i40) and a CSD-17-80-2A-R harmonic drive to amplify the motor torque. A dovetail C-ring mechanism is used to actively support the human upper arm internal/external rotation. Furthermore, a 4 pole EC Maxon motor and a speed reducer drive the C-ring mechanism through a spur gear set.

The elbow joint module consists of a normal revolute joint. A Maxon EC-4 pole motor with a speed reducer located near the elbow joint controls the forearm extension/flexion through a worm gear set. The length of the exoskeleton's upper link is adjustable to adapt the user with different anthropomorphic parameters. Moreover, an upper arm support prevents the offset between the exoskeleton and human anatomical joints, i.e., shoulder and elbow joint, causing an uncomfortable interaction between the two systems. Finally, the wrist module consists of a C-ring mechanism that is designed to support the human wrist rotation (radial/ulnar deviation). A 4 pole EC Maxon motor and a speed reducer located along the forearm likewise actuate the C-ring of the wrist joint.

### 2.2. Kinematics

The kinematic model of the exoskeleton robot is developed by using Denavit–Hartenberg (DH) parameters defined in Table 1, where $L_1$ and $L_2$ represent the lengths of the upper arm and forearm links, respectively. Based on the DH parameters, the transformation matrix is given by

$$T_{i-1,i} = \begin{bmatrix} c\theta_i & -s\theta_i c\alpha & s\theta_i s\alpha_i & a_i c\theta_i \\ s\theta_i & c\theta_i c\alpha_i & -c\theta_i s\alpha_i & a_i s\theta_i \\ 0 & s\alpha_i & c\alpha_i & d_i \\ 0 & 0 & 0 & 1 \end{bmatrix} \tag{1}$$

where $s$ and $c$ represent the sine and the cosine functions, respectively.

The forward kinematics is obtained by computing the overall matrix of transformation from the base frame to the wrist

**Table 1.** Denavit–Hartenberg (DH) parameters.

| Joints | $\alpha_i$ | $a_i$ | $d_i$ | $\theta_i$ |
|--------|-----------|-------|-------|-----------|
| 1 | $\pi/2$ | 0 | 0 | $\pi/2 - \theta_1$ |
| 2 | $\pi/2$ | 0 | $L_1$ | $\pi + \theta_2$ |
| 3 | $-\pi/2$ | 0 | 0 | $\theta_3$ |
| 4 | 0 | 0 | $L_2$ | $\theta_4$ |

$$T_{0,4} = \begin{bmatrix} m_{11} & m_{12} & m_{13} & n_{14} \\ m_{21} & m_{22} & m_{23} & n_{24} \\ m_{31} & m_{32} & m_{33} & n_{34} \\ 0 & 0 & 0 & 1 \end{bmatrix} \tag{2}$$

where all entries are given in the Appendix A.

The inverse kinematics is derived from the transformation matrix (2). The joint angles can be obtained as:

$$\theta_2 = \pi + \arctan 2(n_{34}, n_{14})$$
$$\theta_3 = \pm \arcsin\left(\frac{n_{34}}{L_2 \sin\theta_2}\right)$$
$$\theta_1 = \frac{\pi}{2} + \frac{1}{2}\arcsin\frac{n_{14}-n_{24}}{L_1 + L_2 \cos\theta_3 + L_2 \cos\theta_2 \sin\theta_3}$$
$$\theta_4 = \arccos\left(\frac{m_{32}\cos\theta_2 + m_{31}\cos\theta_3 \sin\theta_2}{\cos^2\theta_2 - \cos\theta_3 \sin\theta_2}\right)$$

(3)

### 2.3. Workspace and Singularity Analysis

The two most important properties that influence the geometrical design of a robotic exoskeleton are workspace and singularity analysis [28]. The kinematic model is used to analyze the workspace of the human upper limb and exoskeleton robot. Given the position of any point in the workspace, it is important to determine whether it belongs to the actual workspace or not, and helps to verify if at least one solution for the joint angles exists [2]. Therefore, a direct search method is employed to essentially evaluate the existence of an inverse kinematics solution for the human and robotic exoskeleton, shown in Figure 2b,c. The kinematic properties selected for this study are given in Appendix C.

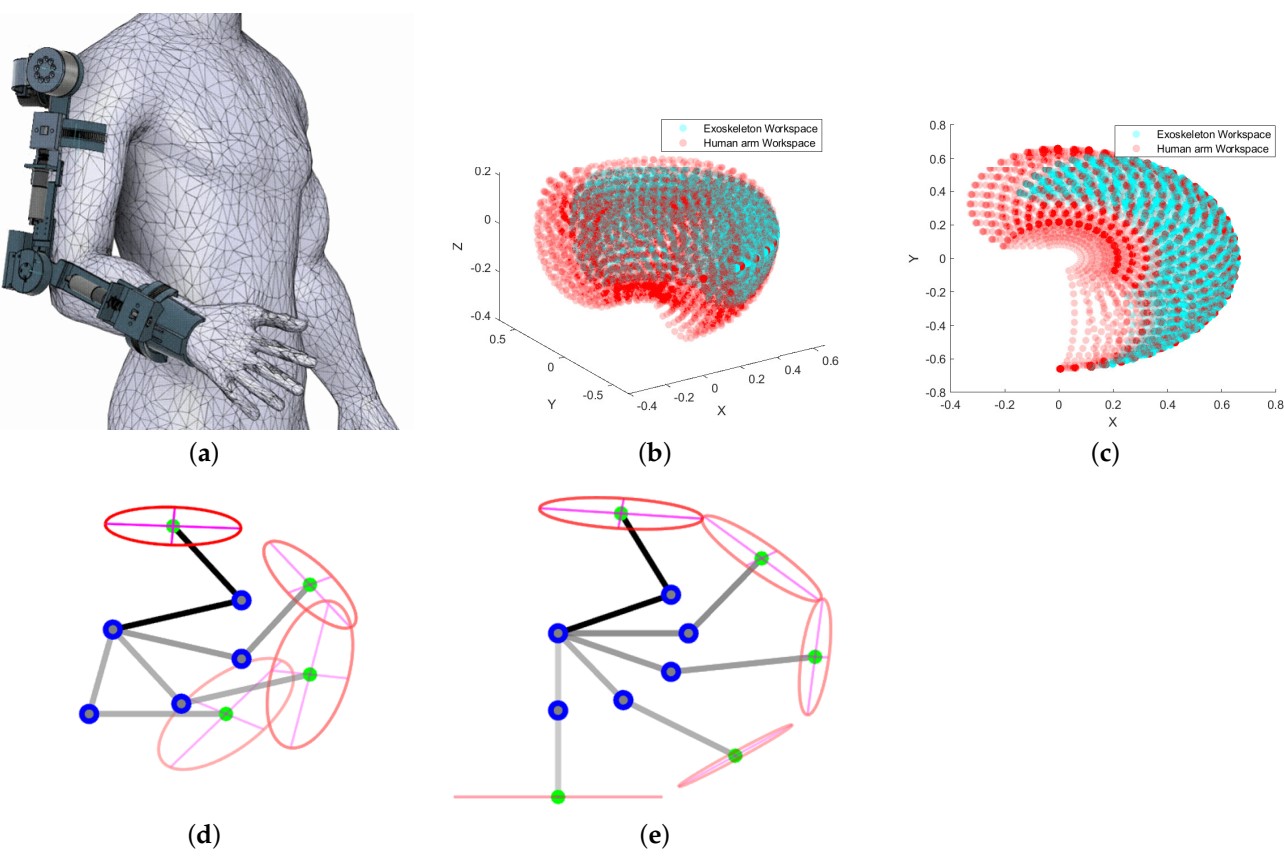

**Figure 2.** Workspace analysis of the 4-DOF upper limb exoskeleton (cyan) within the human arm workspace (red) (measured in meters): (**a**) human–exoskeleton system, (**b**) isotropic view of upper limb exoskeleton and human arm workspace, (**c**) sagittal plane view of upper limb exoskeleton and human arm workspace, (**d**) different configurations of human–exoskeleton system in high manipulability region, (**e**) configurations of human–exoskeleton system in low manipulability region.

Apart from analyzing the reachable workspace, implementation of safe and stable operation is also required due to kinematic singularities within the workspace. Hence, it is necessary to identify all singular configurations while planning trajectories for the robotic exoskeleton. The manipulability ellipsoid and determinant of the Jacobian matrix are the two important indices that characterize the degree of singularity [29]. Our study determines the kinematic performance of the exoskeleton system by analyzing the manipulability

index, which gives us information about the low and high manipulability regions, shown in Figure 2d,e.

In the manipulability analysis, we look at the position of the wrist only. Thus, we take Jacobian in the form of

$$\mathbf{J} = \frac{\partial \boldsymbol{n}}{\partial \boldsymbol{\theta}_i} = \begin{bmatrix} -n_{24} + L_1 c\theta_1 & -n_{34} c\theta_1 & -L_2(s\theta_1 s\theta_3 + c\theta_1 c\theta_2 c\theta_3) \\ n_{14} + L_1 s\theta_1 & -n_{34} s\theta_1 & L_2(c\theta_1 s\theta_3 - s\theta_1 c\theta_2 c\theta_3) \\ 0 & -L_2 c\theta_2 s\theta_3 & -L_2 s\theta_2 c\theta_3 \end{bmatrix} \tag{4}$$

where $\boldsymbol{n} = [n_{14} \; n_{24} \; n_{34}]^T$, $\boldsymbol{\theta}_i = [\theta_1 \; \theta_2 \; \theta_3]^T$. The manipulability index can be determined after computing the Jacobian as follows:

$$\mu(\mathbf{J}) = \sqrt{|\mathbf{J}\mathbf{J}^T|} \tag{5}$$

where $\mu$ is the manipulability index. Figure 2d,e display different configurations of the human upper limb and exoskeleton system and their corresponding manipulability ellipses in high and low region of manipulability. Moreover, the manipulability analysis gives us information about the uniform distribution of the forces and torques applied by the exoskeleton system to the human upper limb [30]. Another important aspect of analyzing the manipulability ellipse is to identify the singular configuration of the exoskeleton system in the workspace. If the determinant of the Jacobian matrix is zero, the robot encounters singularity or exhibit zero manipulability. Hence, this analysis can be used for the robot path planning where it will try to avoid the low region of manipulability.

## 3. Exoskeleton Control System

The control architecture for the upper limb exoskeleton and carbon hand is shown in Figure 3. The control system is implemented in the robotic operating system (ROS), which includes task planning for activities of daily living, a complete path planning for the robotic exoskeleton, computing the inverse kinematics, trajectory generation, and controller design. Furthermore, input control signals are used to perform an ADL while wearing the robotic exoskeleton and carbon hand.

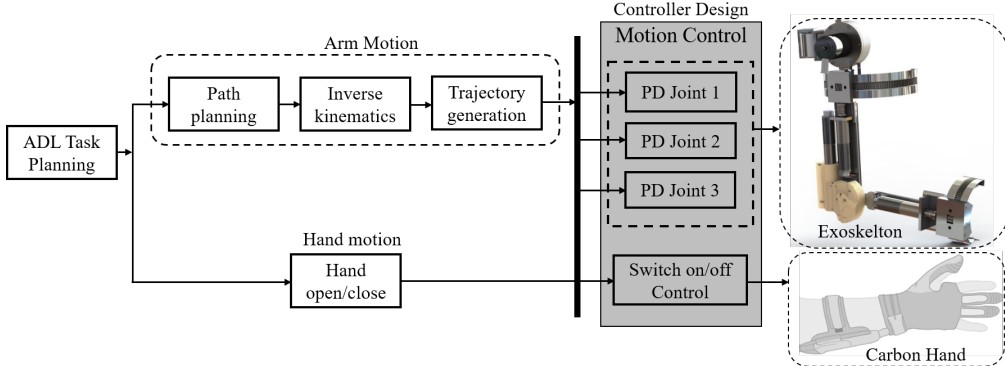

**Figure 3.** An overall system control architecture. The tasks for the activities of daily living (ADL) is predefined. The PD control method is implemented for the individual joint control. A carbon hand developed by SEM glove is adopted to control the hand opening/closing movement (switch on/off control).

The control system consists of four motors, controllers (Maxon EPOS4 Compact 50/8 CAN) and encoders. Moreover, a graphical interface was developed using a combination of PyQt4, Python and ROS that can be used to choose the various types of control modes, tune control parameters, sending high level control commands and real-time logging of data. A CAN bus communication is adopted as the communication method between ROS and Maxon EPOS4.

*PD Control Scheme for Upper Limb Exoskeleton Robot*

The dynamic model of an exoskeleton can be derived using the Lagrange formulation and can be expressed by the following equation

$$M(q)\ddot{q} + C(q,\dot{q})\dot{q} + \tau_g = \tau \tag{6}$$

where $q \in \Re$ is a position vector, $M(q)$ is inertia matrix, $C(q,\dot{q})$ represents the Coriolis forces and $\tau_g$ is the torque due to gravity. Although we have used the model-free PD/PID control scheme, the dynamic model of the system is still provided in (4) to simulate the dynamic response of the robotic exoskeleton. All entries of the dynamic Equation (4) can be found in Appendix B.

In this article, a PD-based trajectory tracking control problem is investigated, where the joint angle trajectories $q$ are bound to track the desired trajectories $q_d$ (Algorithm 1).

The PD control law can be expressed as:

$$
\begin{aligned}
u &= K_p\tilde{q} + K_d\dot{\tilde{q}} \\
\dot{\tilde{q}} &= \dot{q}_d - \dot{q}
\end{aligned}
\tag{7}
$$

where $\tilde{q} = q_d - q$. $K_p$ and $K_d$ are the proportional and differential gains, respectively.

We stabilize the open loop robotic system (6) by using the stability property of the PD control scheme (9) and form a stable closed loop system as follows:

$$
\begin{aligned}
M(q)\ddot{q} + C(q,\dot{q})\dot{q} + \tau_g &= K_p\tilde{q} + K_d\dot{\tilde{q}} \\
\dot{\tilde{q}} &= \dot{q}_d - \dot{q}
\end{aligned}
\tag{8}
$$

The equation can be written in the matrix form as:

$$
\begin{bmatrix} \dot{\tilde{q}} \\ \ddot{\tilde{q}} \end{bmatrix} = \begin{bmatrix} \dot{q}_d - \dot{q} \\ \ddot{q}_d + \frac{1}{M}(C\dot{q} + g - K_p\tilde{q} - K_d\dot{\tilde{q}}) \end{bmatrix} \qquad \because \ddot{q} = \ddot{q}_d - \ddot{\tilde{q}}
\tag{9}
$$

---

**Algorithm 1** PD-based trajectory tracking for each joint

---

**Given:**
- Sampling time: $T_s$
- User define parameters: $k_p$, $k_d$
- Desired trajectory: $q_d(k)$

**Initialization:**
- $\tilde{q}(0) = 0$
- $k \leftarrow 0$

**Repeat:**
- $\tilde{q}(k) = q_d(k) - q(k)$, $\dot{\tilde{q}}(k) = \frac{\tilde{q}(k) - \tilde{q}(k-1)}{T_s}$
- $Output = k_p\tilde{q}(k) + k_d\dot{\tilde{q}}(k)$
- $\tilde{q}(k-1) \leftarrow \tilde{q}(k)$
- $k \leftarrow k + 1$

---

## 4. Control Implementation in the Upper Limb Exoskeleton and Experimental Evaluation

The challenge of the human–exoskeleton system lies in its complicated interaction in which the robotic motion is coupled with the human upper limb musculoskeletal system. Thus, we have selected joint angle trajectories to evaluate the system's performance, which helps us to analyze the influence of the kinematic/kinematic properties of the human–exoskeleton system for different manipulation activities. In this section, a model-free PD-based trajectory tracking is implemented to demonstrate the performance of the wheelchair exoskeleton. The architecture of the control scheme is presented in Figure 3.

In our study, we selected two tasks to evaluate the effectiveness of using a wheelchair exoskeleton, shown in Figures 4 and 5. Several positions in the task space were preliminarily

defined via human demonstration, and the trajectories were generated in the joint space corresponding to each task. Sixteen trials were recorded from the two subjects for each task, where they were instructed to sit in the wheelchair by wearing the exoskeleton and forced to follow the desired joint angle trajectories, shown in Figure 6. The joint angle trajectories were recorded, and the whole system was evaluated upon the tracking performance of all joints represented by the root mean square (RMSE) value from the 16 trials shown in Figure 7. The detailed statistics representing the human–exoskeleton system's performance are listed in Table 2. For the normal drinking task, it was noted that the human–exoskeleton system was able to satisfactorily track the reference trajectories and shown average RMSE values of 0.0247 rad, 0.0210 rad, and 0.0207 rad for the three joints, respectively. Moreover, the variation in the RMSE values among the 16 trials was also in the acceptable range, i.e., 0.0184 rad, 0.0027 rad, and 0.0071 rad for all three joints, which shows that the human–exoskeleton was able to perform the task during different trials satisfactorily.



**Figure 4.** Demonstration of normal drinking task: (**a**) initial position $t = 0$ s, (**b**) exoskeleton moves to the grasping position $t = 11$ s, (**c**) the drinking position $t = 31$ s, (**d**) drop the bottle to the table $t = 51$ s, (**e**) get back to the initial position $t = 72$ s.

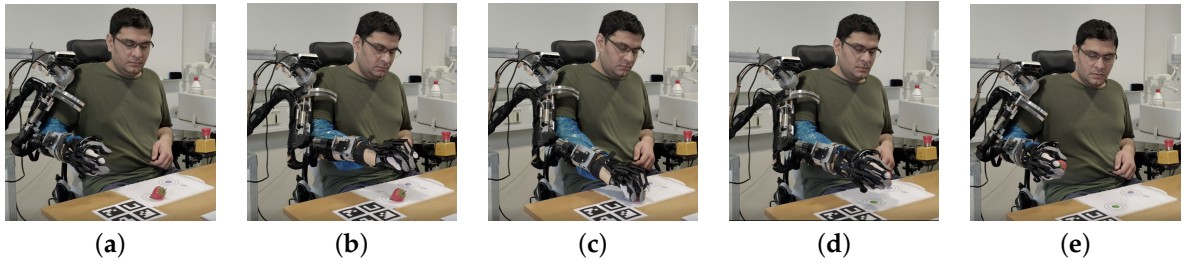

**Figure 5.** Demonstration of an object picking-up task: (**a**) initial position $t = 0$ s, (**b**) moving over the target $t = 8$ s, (**c**) exoskeleton moves to a grasping position and hold the object $t = 16$ s, (**d**) pick the object up $t = 24$ s, (**e**) get back to the initial position $t = 42$ s.

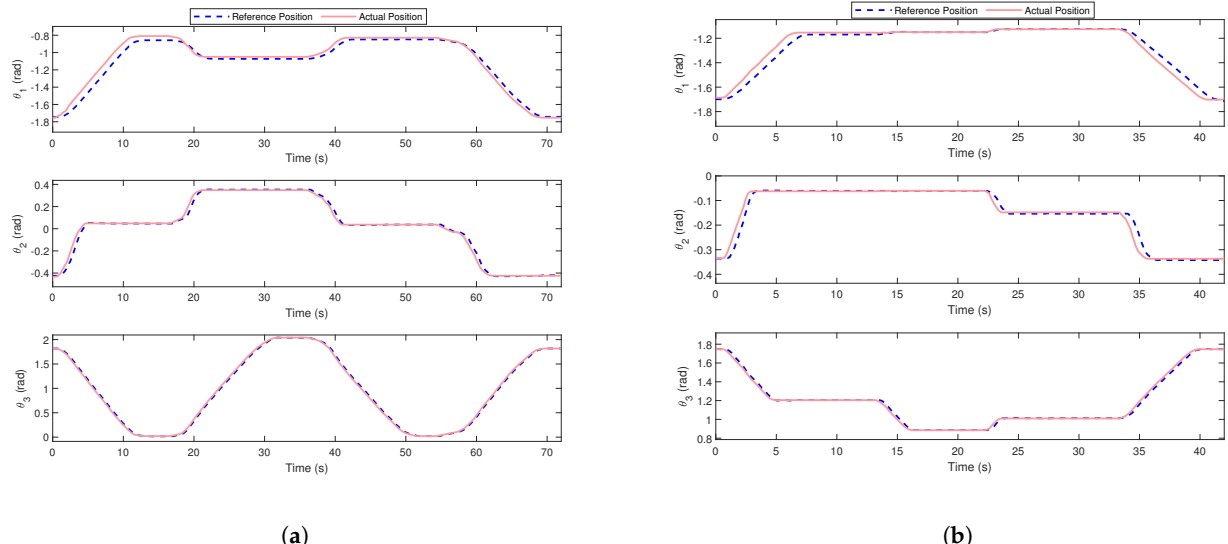

**Figure 6.** Exoskeleton's trajectory tracking control performance assessment for two different ADL. (**a**) PD-based trajectory tracking controls for drinking task. (**b**) PD-based trajectory tracking control for an object picking-up task.

**Table 2.** Statistical analysis of PD control method using RMSE value for the performance assessment of the wheelchair exoskeleton.

| Joints | Average RMSE | Max RMSE | Min RMSE | Variance of RMSE | Average RMSE | Max RMSE | Min RMSE | Variance RMSE |
|---|---|---|---|---|---|---|---|---|
| | | Drinking Task | | | | Object Picking Task | | |
| Joint 1 | 0.0247 | 0.0382 | 0.0198 | 0.0184 | 0.0360 | 0.0402 | 0.0323 | 0.0079 |
| Joint 2 | 0.0210 | 0.0223 | 0.0196 | 0.0027 | 0.0146 | 0.0155 | 0.0131 | 0.0024 |
| Joint 3 | 0.0207 | 0.0238 | 0.0167 | 0.0071 | 0.0184 | 0.0213 | 0.0150 | 0.0062 |

The robotic system and the control algorithms can be designed to fulfil the requirement for a particular task, but sometimes it is hard to achieve generality. Thus, to maximize the functional reliability of the presented system, we selected a second task to evaluate the system's performance, shown in Figure 5. It is noted that the tracking performance of the shoulder joint was reduced, while the tracking accuracy of Joint 2 and Joint 3 was increased compared to the normal drinking task, shown in Figure 7. In general, the variation in the RMSE values among the 16 trials was in an acceptable range, i.e., 0.0079 rad, 0.0024 rad, and 0.0062 rad for all three joints.

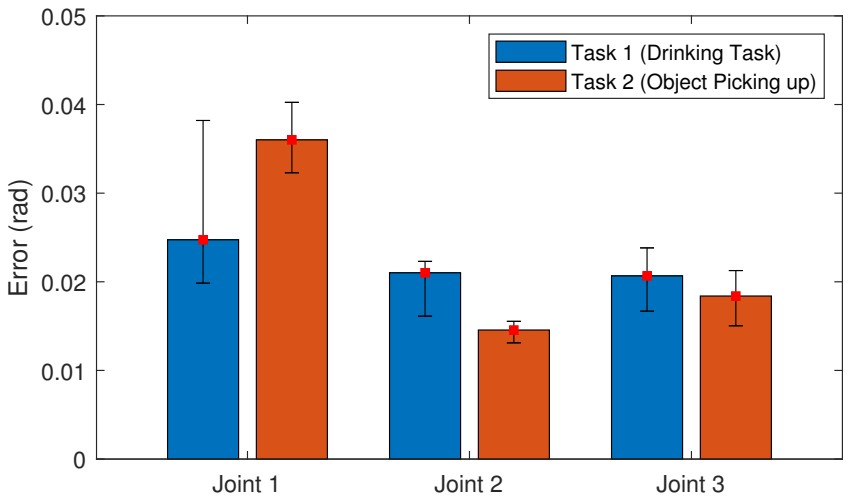

**Figure 7.** Bar diagram of the RMSE with variance from 16 trials for each task.

## 5. Discussion

Table 2 summarizes the results of two experiments and presents a statistical analysis of the joint trajectories to demonstrate the effectiveness of using an exoskeleton system for basic ADLs. The data illustrated in Figure 7 show the variations in mean RMSE values among two tasks in joint space. Several parameters, such as mechanism design, mode of actuation, selection of control method to accommodate variations in payload, and human anthropomorphic parameters, may influence the functional reliability of the human–exoskeleton system.

Implementation of a basic PD control method without gravity compensation/human arm weight compensation and a backdrivability of shoulder mechanism had made it difficult to achieve a more precise control compared to the other joint mechanisms. Therefore, average RMSE values for the shoulder joint were comparatively higher than the other two joints. Teng et al. [1] implemented a PD control with gravity compensation and analyzed the effect of uncertain dynamics and external disturbances on the human–exoskeleton system. Data presented in [1,31,32] have shown that the performance of an exoskeleton driving human shoulder joint was comparatively lower than the elbow joint exoskeleton because the gravity torque induced by variable payload and human arm weight may affect

the relative precision. Alternatively, the C-ring and worm gear mechanisms responsible for supporting a human shoulder joint rotation and elbow joint movements take advantage of a large reduction ratio. The design also facilitates holding the output position without energy consumption because of its non-backdrivability. Moreover, the two joints are relatively less affected by the variation in the payload and upper limb anatomy as can be seen from error bar diagram presented in Figure 7. Deyby et al. [12] presented a similar mechanism and analyzed the position and orientation synchronization between the human upper limb and exoskeleton.

The study demonstrates that the exoskeleton presented is applicable for motion assistance of physically impaired people in their ADLs. In our future work, we will extend this study to clinically evaluate the system to examine comfort, patient acceptance, and functional use of the system with severe to moderate upper limb impairment. We will look into more advanced control methods to compensate for ill effects caused by uncertain dynamics and external disturbances and study their implications for assistive applications. Manipulability/singularity free workspace is another important factor that should be considered during the path planning of an exoskeleton robot. Future work will focus on developing a method to optimize the exoskeleton's trajectory in the task space and attempt to maximize likelihood of manipulation in the high manipulability region; thereby, it guarantees uniform distribution of forces and the torques and improves the physical human–robot interaction.

## 6. Conclusions

In this article, we present the mechanical design, control, and performance evaluation of the wheelchair exoskeleton for physical assistance. The design takes advantage of non-backdrivable mechanisms and holds the output position of the exoskeleton without energy consumption. Furthermore, an overall structure of the exoskeleton system offers compatible kinematics and provides a safer ROM that generates a variety of unconstrained motions for active assistance.

The experiments performed evaluated the system's response to shoulder extension/flexion, shoulder internal/external rotation, and elbow extension/flexion for two different ADLs. The statistical analysis of the joint angle trajectories shows that the proposed system and the implementation of PD-based control method are appropriate for performing several essential tasks. Upon the data presented, it is expected that the system will be able to support the tetraplegia users in different ADLs, such as drinking/eating, which helps them in maintaining an independent lifestyle.

**Supplementary Materials:** The following are available at https://www.mdpi.com/article/10.3390/app11135865/s1.

**Author Contributions:** Conceptualization, M.A.G., S.B., M.T. and L.N.S.A.S.; methodology, M.A.G.; software, M.T., S.H.B. and M.M.; validation, M.A.G.; formal analysis, M.A.G.; investigation, M.A.G.; writing—original draft preparation, M.A.G.; writing—review and editing, M.A.G., S.B., M.T., L.N.S.A.S. and T.B.M.; supervision, S.B. and T.B.M., T.B. All authors have read and agreed to the published version of the manuscript.

**Funding:** This work has been supported by AAU EXOTIC project.

**Institutional Review Board Statement:** Not applicable.

**Informed Consent Statement:** The experiments presented in this study were the part of system testing thus did not require consent statement.

**Data Availability Statement:** Data is contained within the article.

**Conflicts of Interest:** The authors declare no conflict of interest.

## Appendix A

$$m_{11} = c\theta_4(s\theta_1 s\theta_3 + c\theta_1 c\theta_2 c\theta_3) - c\theta_1 s\theta_2 s\theta_4$$

$$m_{12} = -s\theta_4(s\theta_1 s\theta_3 + c\theta_1 c\theta_2 c\theta_3) - c\theta_1 c\theta_4 s\theta_2$$

$$m_{13} = c\theta_3 s\theta_1 - c\theta_1 c\theta_2 s\theta_3$$

$$m_{21} = -c\theta_4(c\theta_1 s\theta_3 - c\theta_2 c\theta_3 s\theta_1) - s\theta_1 s\theta_2 s\theta_4$$

$$m_{22} = s\theta_4(c\theta_1 s\theta_3 - c\theta_2 c\theta_3 s\theta_1) - c\theta_4 s\theta_1 s\theta_2$$

$$m_{23} = -c\theta_1 c\theta_3 - c\theta_2 s\theta_1 s\theta_3$$

$$m_{31} = c\theta_2 s\theta_4 + c\theta_3 c\theta_4 s\theta_2$$

$$m_{32} = c\theta_2 c\theta_4 - c\theta_3 s\theta_2 s\theta_4$$

$$m_{33} = -s\theta_2 s\theta_3$$

$$n_{14} = L_2(c\theta_3 s\theta_1 - c\theta_1 c\theta_2 s\theta_3) + L_1 s\theta_1$$

$$n_{24} = -L_2(c\theta_1 c\theta_3 + c\theta_2 s\theta_1 s\theta_3) - L_1 c\theta_1$$

$$n_{34} = -L_2 s\theta_2 s\theta_3$$

## Appendix B

$$M_{11} = I_1 + L_1^2 m_3 + L_{c1}^2 m_1 + L_{c1}^2 m_2 + L_{c2}^2 m_3 c\theta_2^2 + L_{c2}^2 m_3 c\theta_3^2 + 2L_1 L_{c2} m_3 c\theta_3 - L_{c2}^2 m_3 c\theta_2^2 c\theta_3^2$$

$$M_{12} = L_{c2} m_3 s\theta_2 s\theta_3 (L_1 + L_{c2} c\theta_3)$$

$$M_{13} = -L_{c2} m_3 c\theta_2 (L_{c2} + L_1 c\theta_3)$$

$$M_{21} = L_{c2} m_3 s\theta_2 s\theta_3 (L_1 + L_{c2} c\theta_3)$$

$$M_{22} = -m_3 L_{c2}^2 c\theta_3^2 + m_3 L_{c2}^2 + I_2$$

$$M_{23} = 0$$

$$M_{31} = -L_{c2} m_3 c\theta_2 (L_{c2} + L_1 c\theta_3)$$

$$M_{32} = 0$$

$$M_{33} = m_3 L_{c2}^2 + I_3$$

$$C_1 = L_{c2} m_3 (L_1 \dot{\theta}_2^2 c\theta_2 s\theta_3 + L_1 \dot{\theta}_3^2 c\theta_2 s\theta_3 - 2L_1 \dot{\theta}_1 \dot{\theta}_3 s\theta_3 - L_{c2} \dot{\theta}_1 \dot{\theta}_2 s2\theta_2 - L_{c2} \dot{\theta}_1 \dot{\theta}_3 s2\theta_3$$
$$+ L_{c2} \dot{\theta}_2^2 c\theta_2 c\theta_3 s\theta_3 + 2L_1 \dot{\theta}_2 \dot{\theta}_3 c\theta_3 s\theta_2 + 2L_{c2} \dot{\theta}_2 \dot{\theta}_3 c\theta_3^2 s\theta_2 + 2L_{c2} \dot{\theta}_1 \dot{\theta}_2 c\theta_2 c\theta_3^2 s\theta_2 + 2L_{c2} \dot{\theta}_1 \dot{\theta}_3 c\theta_2^2 c\theta_3 s\theta_3)$$
$$C_2 = (L_{c2}^2 m_3(-2c\theta_2 s\theta_2 \dot{\theta}_1^2 c\theta_3^2 + s2\theta_2 \dot{\theta}_1^2 + 4\dot{\theta}_3 s\theta_2 \dot{\theta}_1 c\theta_3^2 - 4\dot{\theta}_3 sin\theta_2 \dot{\theta}_1 + 2\dot{\theta}_2 \dot{\theta}_3 sin2\theta_3))/2$$
$$C_3 = (L_{c2} m_3(2L_1 \dot{\theta}_1^2 s\theta_3 + L_{c2} \dot{\theta}_1^2 sin2\theta_3 - L_{c2} \dot{\theta}_2^2 s2\theta_3 + 4L_{c2} \dot{\theta}_1 \dot{\theta}_2 s\theta_2 - 2L_{c2} \dot{\theta}_1^2 c\theta_2^2 c\theta_3 s\theta_3 - 4L_{c2} \dot{\theta}_1 \dot{\theta}_2 c\theta_3^2 s\theta_2))/2$$

$$G_1 = L_1 m_3 s\theta_1 + L_{c1} m_1 s\theta_1 + L_{c1} m_2 s\theta_1 + L_{c2} m_3 c\theta_3 s\theta_1 - L_{c2} m_3 c\theta_1 c\theta_2 s\theta_3$$

$$G_2 = L_{c2} m_3 s\theta_1 s\theta_2 s\theta_3$$

$$G_3 = L_{c2}m_3(c\theta_1 s\theta_3 - c\theta_2 c\theta_3 s\theta_1)$$

$L_{c1}$ is the distance of the center of mass of the exoskeleton's upper arm from the shoulder joint, and $L_{c2}$ is the distance of the center of mass of the exoskeleton's forearm from the elbow joint. The parametric values for the above dynamic system (upper limb exoskeleton) are assumed to be: $m_1 = 2.5$ kg, $L_1 = 0.33$ m, $I_1 = 0.20$ kg m$^2$, $m_2 = 1.5$ kg, $L_2 = 0.246$ m, $I_2 = 0.15$ kg m$^2$.

## Appendix C

**Table A1.** Mechanical properties of the exoskeleton, and the average estimated anthropomorphic parameters for human subjects.

| Link | Exoskeleton | | Human Subject | |
|------|-------------|-----------|---------------|-----------|
| | Length (m) | Weight (kg) | Length (m) | Weight (kg) |
| Upper arm | 0.33 | 2.5 | 0.33 | 1.386 |
| Forearm | 0.246 | 1.5 | 0.37 | 0.886 |

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
