# Peer review of "A 4-DOF Upper Limb Exoskeleton for Physical Assistance: Design, Modeling, Control and Performance Evaluation"

_applsci, doi:10.3390/app11135865_

Round 1

Reviewer 1 Report

The authors have presented a 4-DOF exoskeleton device for physical assistance. The architecture, kinematics and control strategies were well presented. A working video of the experiment could be more informative to the readers.

Author Response

We thank the reviewer for his recognition of this work. Upon the reviewer’s recommendation, we have included a small working video in which human is approaching toward the bottle.

Reviewer 2 Report

This paper presents a 4 DOF mechanism to be used for the movement of the upper limb. Both mechanical and control strategies are outlined.

The paper is rather complete, but to me, there are some grey areas to be addressed:

  1. In the Introduction section authors have focused on shoulder mechanisms, not considering both elbow and wrist mechanisms (which have been widely studied). Please improve this aspect. An example is [1]
  2. Section 2 needs to be improved. In particular, it may be clear for a person which is used to Denavit-Hartemberg, but may not be clear for other researchers. In referring to:
    • Line 122: "transformation matrix" is related to which transformation?
    • If the wrist is "least affected", why theta4 as not been calculated?
    • Eq (4): the directions of which Jacobian J is related to should be stated (x-y-rotation? x-y-z?)
  3. In Section 3 it is not necessary to state in two adjacent phrases that the control system is implemented in ROS.
  4. Eq (6): why no stiffness has been considered?
  5. Eq (7): if qtilde = q-qd, why its derivative dqtilte is -dq and not dq-dqd? And then why dqtilde has been substituted in Eq (9) with dqtilte_d?

[1] Zuccon, G.; Bottin, M.; Ceccarelli, M.; Rosati, G. Design and Performance of an Elbow Assisting Mechanism. Machines 2020, 8, 68. https://doi.org/10.3390/machines8040068

Author Response

We thank the reviewer for his appreciation and recognition of this work. We have attempted to address all the comments.

Response to the Comment 1:

We have included review on elbow and wrist by referring to the suggested references and others. Please refer to the highlighted text in Section 1 (Introduction), Paragraph 1.

Response to the Comment 2:

  • The transformation matrix shows the orientation and position of the end-effector with respect to the base frame. Please refer to the highlighted text in Section 2.2 (Kinematics).

  • We have included q4 in inverse kinematics. And also describe that why we choose to ignore it in our study. Please refer to the Section 2.2 (Kinematics)

  • The Jacobian is associated with the shoulder joint  and elbow joint. Please see the highlighted text in Section 2.3 (Workspace and singularity analysis), Paragraph 2.

Response to the Comment 3:

Corrected. Please refer to the highlighted text in Section 3, Paragraph 1.

Response to the Comment 4:

Since we do not use human-robot interaction and the presented work has only considered the mechanical model of the system. Therefore, we did not considered the stiffness.

Response to the Comment 5:

Thanks for the identification of this glitch. We have carefully revised and corrected the equations. Please refer to the highlighted text and equations in Section 3.1 (PD control scheme for upper limb exoskeleton robot)

Reviewer 3 Report

The research focuses on the development of an exoskeleton and the implementation of an approach for motion assistance of physically impaired people in their ADLs. The work is well motivated.

A system of deep mathematical methods was involved. It was demonstrated that it is possible to execute various tasks by evaluating tracking performance.

A really great job has been done.

I have my own ideas, but they relate only to my vision of the problem and its solution, so I will refrain from criticism and recommend the work for publication in its present form.

Author Response

We thank the reviewer for his appreciation and recognition of this work.

Reviewer 4 Report

The paper introduces an upper limb exoskeleton to be used in combination with wheelchairs.

The paper is interesting and it is well structured in terms of formal analysis and preliminary results.

However the introduction should be improved detailing better how and which are the important parameters for arm rehabilitation exercises, these can be done referencing and using this works that explain in detail these issues:

Chaparro-Rico, B.D.M., Cafolla, D., Castillo-Castaneda, E., Ceccarelli, M. Design of arm exercises for rehabilitation assistance (2020) Journal of Engineering Research (Kuwait), 8 (3), pp. 203-218. DOI: 10.36909/JER.V8I3.6523

Chaparro-Rico, B., Cafolla, D., Ceccarelli, M., Castillo-Castaneda, E. Design and simulation of an assisting mechanism for arm exercises (2017) Mechanisms and Machine Science, 47, pp. 115-123. DOI: 10.1007/978-3-319-48375-7_13

Figure 6 should be improved enlarging the detail since numbers are hard to see an trend too.

Ethical approvement is missing togheter with the document number and date of approval. This is an important lack since the experiment are done using a person embraced by an apparatus. Please provide it.

Author Response

The authors would like to thank the reviewer for their time spent reviewing our article and helping us by providing his/her valuable /constructive comments. The suggestions from the referee were taken into consideration, and the paper has been revised carefully.

Reviewer's Comment 1:

However the introduction should be improved detailing better how and which are the important parameters for arm rehabilitation exercises, these can be done referencing and using this works that explain in detail these issues:

We have revised Introduction by referring more state-of-the-art, including the suggested papers.  Please refer to the highlighted text in Section 1 (Introduction), Paragraph 1 and Paragraph 2.

Reviewer's Comment 2:

Figure 6 should be improved enlarging the detail since numbers are hard to see an trend too.

Done.

Reviewer's Comment 3:

Ethical approvement is missing together with the document number and date of approval. This is an important lack since the experiment are done using a person embraced by an apparatus. Please provide it.

The exoskeleton is itself has been approved for use on healthy and disabled individuals for performance of similar task in an ethical approved experimental study. On the other hand, the protocol, the system control testing is not particularly specified in the protocol. Thanks for this suggestion. We are working on get clarification for this matter.